

# Examining the implications of photochemical indicators on the O₃-NOₓ-VOC sensitivity and control strategies: A case study in the Yangtze River Delta (YRD), China

Xun Li[1], Momei Qin[1], Lin Li[1], Kangjia Gong[1], Huizhong Shen[2], Jingyi Li[1], Jianlin Hu[1]

[1]Jiangsu Key Laboratory of Atmospheric Environment Monitoring and Pollution Control, Collaborative Innovation Center of Atmospheric Environment and Equipment Technology, Nanjing University of Information Science and Technology, Nanjing 210044, China

[2]School of Environmental Sciences and Engineering, Southern University of Science and Technology, Shenzhen, 518055, China

*Correspondence to*: Momei Qin (momei.qin@nuist.edu.cn); Jianlin Hu (jianlinhu@nuist.edu.cn)

**Abstract.** Ozone ($O_3$) has become a significant air pollutant in China in recent years. $O_3$ abatement is challenging due to the nonlinear response of $O_3$ to precursors nitrogen oxides ($NO_x$) and volatile organic compounds (VOCs). Photochemical indicators are widely used to estimate the $O_3$-$NO_x$-VOC sensitivity, and this has important policy implications. However, the effectiveness of the indicators has seldom been evaluated. This study examines the applications of four indicators that include the ratio of the production rates of $H_2O_2$ and $HNO_3$ ($P_{H2O2}/P_{HNO3}$), $HCHO/NO_2$, $HCHO/NO_y$, and reactive nitrogen ($NO_y$) in the Yangtze River Delta (YRD) with localized thresholds. The overall accuracy was high (> 92%) for all indicators and not significantly reduced with different simulation periods or in different locations of the region. By comparing with the $O_3$ isopleths, it was found that $HCHO/NO_2$ and $HCHO/NO_y$ showed the most consistency, whereas $P_{H2O2}/P_{HNO3}$ ($NO_y$) tended to underestimate (overestimate) the positive response of $O_3$ to $NO_x$. Additionally, $P_{H2O2}/P_{HNO3}$ was less likely to attribute the $O_3$ formation to mixed sensitivity than the other indicators, and this demonstrated a preference for a single-pollutant control strategy. This study also revealed that the details in the methodology used to derive the threshold values impacted the results, and this may produce uncertainties in the application of photochemical indicators.

## 1 Introduction

Ozone ($O_3$) is one of the most important pollutants in the atmosphere, high concentrations of which pose a serious threat to human health, ecosystems, and global climate change (Wang et al., 2020b; Liang et al., 2019; De Marco et al., 2022; Feng et al., 2022; Skeie et al., 2020). Many metropolitan areas in China, such as Beijing-Tianjin-Hebei (BTH), the Yangtze River Delta (YRD), and Pearl River Delta (PRD), have been experiencing severe $O_3$ pollution in recent years (Gao et al., 2017; Lu et al., 2018). Moreover, unlike the decline in fine particulate matter ($PM_{2.5}$), the $O_3$ concentrations in urban areas have



increased (Wang et al., 2019; Lu et al., 2020; Li et al., 2020).


$O_3$ is produced by photochemical reactions involving nitrogen oxides ($NO_x$) and volatile organic compounds (VOCs). However, the response of $O_3$ to either precursor is nonlinear due to the changing roles of $NO_x$ and VOCs with different VOC/$NO_x$ ratios in $O_3$ chemistry, making $O_3$ pollution abatement more challenging (Wang et al., 2011; Liu et al., 2013). Briefly, when $O_3$ increases with a reduction (increase) in $NO_x$ (VOCs), $O_3$ formation is in a $NO_x$-saturated or VOC-limited

regime that tends to occur in urban areas or cold seasons with relatively lower VOC/$NO_x$ ratios (Tan et al., 2018; Sillman, 1995). Conversely, $O_3$ decreases with reduced $NO_x$ and is typically insensitive to VOC levels in a $NO_x$-limited regime. This is often the case in rural and remote regions with lower $NO_x$ emissions (Murphy et al., 2007; Sillman, 1999). Importantly, a reduction in either $NO_x$ or VOCs can lead to less $O_3$ production when the $O_3$ formation regime is shifted from one to the other, and this is referred to as the transitional regime (Chen et al., 2021; Jin and Holloway, 2015).


Many approaches, including photochemical indicators, the observation-based modeling method (OBM), and the emission-based method (EBM) have been developed to predict the $O_3$-$NO_x$-VOC sensitivity (Sillman, 2002; Wang et al., 2020a; Shen et al., 2021). Among these approaches, the values of the photochemical indicators (such as VOC/$NO_x$, the ratios of hydrogen peroxide to nitric acid ($H_2O_2$/$HNO_3$) or production rates of $H_2O_2$ and $HNO_3$ ($P_{H2O2}$/$P_{HNO3}$), the ratios of formaldehyde to

nitrogen dioxide or reactive nitrogen (HCHO/$NO_2$ or HCHO/$NO_y$), and the $O_3$ to nitric acid ratios ($O_3$/$HNO_3$), $NO_y$) can be derived directly from ground-based measurements, chemical transport models, or even satellite measurements if available, and thus have been widely used in previous studies (Martin et al., 2004; Jin and Holloway, 2015; Jiménez and Baldasano, 2004; Sillman, 2002; Liu et al., 2010). For instance, Ye et al. (2021) presented hourly $H_2O_2$/$NO_z$ ratios based on the measurements at Mount Tai and found that $O_3$ formation is VOC-limited in the morning ($H_2O_2$/$NO_z$<0.15) and switched to

$NO_x$-limited ($H_2O_2$/$NO_z$>0.2) in the afternoon. The ratio $H_2O_2$/$HNO_3$ is considered one of the most robust indicators to determine $O_3$ sensitivity, while the thresholds are highly variable for different regions (Peng et al., 2011; Xie et al., 2014; Ye et al., 2016). $P_{H2O2}$/$P_{HNO3}$ is primarily used in modeling work, such as the spatial distribution of $O_3$ formation regimes over the modeling domain that was exhibited with the indicator (Liu et al., 2010; Du et al., 2022; Zhang et al., 2020). Based on long-term measurements of HCHO and $NO_2$ columns from satellites, Jin and Holloway (2015) found that near-surface $O_3$

formation in several megacities in China had changed from VOC-limited in 2005 to transitional in 2013 with reduced $NO_2$.

The thresholds of photochemical indicators that split VOC-limited, transitional, and $NO_x$-limited regimes vary from region to region, largely depending on local emissions and possibly related to meteorological conditions as well. It was found that the primary emissions of HCHO increased the value of HCHO/$NO_2$, and therefore, the $O_3$ sensitivity determined by the ratio could have been biased with given thresholds. For example, some VOC-limited regimes are misclassified as transitional (Liu



et al., 2021). Du et al. (2022) compared the spatial distribution of $O_3$ sensitivity regimes diagnosed using different indicators and pointed out that $P_{H2O2}/P_{HNO3}$ could be more affected by local emissions than $HCHO/NO_2$. As $H_2O_2$ and $HNO_3$ are very soluble and susceptible to deposition and aerosol formation, the indicator values that involve the two species may change with the meteorological conditions (Castell et al., 2009). However, certain thresholds were often used without considering
the variability, such as the range of one to two for $HCHO/NO_2$ (Jin and Holloway, 2015; Tang et al., 2012; Ma et al., 2021). In particular, the integrated source apportionment method (ISAM) implemented in the community multiscale air quality model (CMAQ) attributes $O_3$ production to either VOC or $NO_x$ tracers based on a comparison of the instantaneous $P_{H2O2}/P_{HNO3}$ with 0.35, and this ultimately affects the source apportionment of $O_3$ using ISAM (Kwok et al., 2015). Whether photochemical indicators can accurately predict $O_3$ sensitivity to precursors with given threshold values has seldom been
examined.

In light of the above, this work aims to revisit the effectiveness of photochemical indicators in the prediction of the $O_3$-precursor response based on a case study in the YRD region in Eastern China. Specifically, the work deployed the mostly-used method to derive localized thresholds of the four photochemical indicators (i.e., $P_{H2O2}/P_{HNO3}$, $HCHO/NO_2$, $HCHO/NO_y$,
and $NO_y$) for Jiangsu Province in the YRD that has been experiencing severe photochemical pollution (Qin et al., 2021; Xu et al., 2021; Lu et al., 2018). The assessment was conducted to examine three aspects: (1) whether the localized threshold values are appropriate for a different location or a different year; (2) the consistency of the photochemical indicator approach with $O_3$ isopleths plots for predicting the $O_3$-$NO_x$-VOC response; and (3) possible uncertainties induced by the methodology with which the threshold values are determined. This work can provide insight into the use of photochemical indicators in
understanding $O_3$ formation and making effective strategies for emission control.

## 2 Methods

### 2.1 Model configurations

The CMAQ model version 5.2 was applied to simulate the photochemical pollution over the YRD and its surrounding areas (Fig. 1) in July 2017. The simulation was performed at a 4-km resolution, using the chemical mechanism of saprc07tic. A
regional $O_3$ pollution event that occurred from July 22 to 31 with the mean level of the observed daily maximum 8-hr average (MDA8) $O_3$ exceeding 85 ppb over the YRD was selected to study the relationship between the $O_3$-$NO_x$-VOC sensitivity and the values of the photochemical indicators. The CMAQ model was driven by a mesoscale meteorological model, the Weather Research and Forecasting (WRF) version 4.2.2. The 2017 emission inventory of anthropogenic air pollutants in the YRD compiled by the Shanghai Academy of Environmental Sciences (An et al., 2020) was adopted, with
the details of the other emissions (i.e., biogenic emissions and open burning) found in Hu et al. (2016).

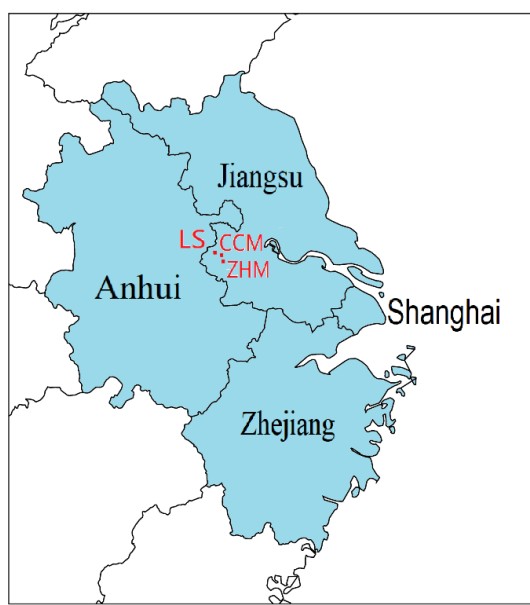

**Figure 1: The simulation domain. The blue area indicates the YRD including Jiangsu, Anhui, Shanghai, and Zhejiang. Three representative sites of Caochangmen (CCM), Zhonghuamen (ZHM), and Laoshan (LS) are marked.**


## 2.2 Model performance evaluation

The model performance on $O_3$ was examined by comparing the simulation with observational data at the monitoring sites supervised by the China National Environmental Monitoring Center (CNEMC) in 13 cities in Jiangsu Province. Three metrics (the normalized mean bias (NMB), the normalized mean error (NME), and the correlation coefficient (r)) as in Sheng et al. (2022) were calculated to evaluate the agreement between the simulation and observations. The criteria of the model performance were recommended by Emery et al. (2017). Table S1 shows that the NMB (NME) values in 12 (10) out of 13 cities met the criteria, indicating that the overall performance was good. In general, the model overestimated $O_3$, with the NMB in the range of -2%−17%. The correlations of the simulation with the observations (all r values above 0.5) indicated that hourly variations in the simulation agree well with the observations. Additionally, three sites in Nanjing (a megacity in Jiangsu), including Caochangmen (CCM), Zhonghuamen (ZHM), and Laoshan (LS), representing downtown, upwind, and downwind areas of the city (see Fig. 1), respectively, were selected to examine the consistency between the photochemical indicators and the $O_3$ isopleths in the implication of $O_3$-precursor sensitivities. Figure S1 shows that the model well reproduced the $O_3$ pollution events at the three sites during the simulation period.



Other species involved in the photochemical indicators, such as HCHO, HNO$_3$, NO$_2$, and NO$_y$ (NO$_y$ (g)=NO+NO$_2$+NO$_3$+2×N$_2$O$_5$+HONO+HNO$_3$+HNO$_4$+peroxyacetyl nitrate and its homologs (PANs)+alkyl nitrates (ANs) and nitrogen oxychlorides (ClNO$_x$) in the model), were examined and compared with observations during the EXPLORE-YRD campaign (Li et al., 2021). Figure S2 showed that HCHO was underestimated by 51%, while NO$_2$ and NO$_y$ were slightly overestimated, with NMBs of 16% and 7%, respectively. The anthropogenic emission inventory and model were set

up as in the simulation for the EXPLORE-YRD campaign, and the simulated ratios of HCHO/NO$_2$ and HCHO/NO$_y$ in this study could have been underestimated. In addition, the simulation could represent an environment with less abundant VOCs than in the real atmosphere over the YRD.

**2.3 Determination of the indicator thresholds**

The use of photochemical indicators requires threshold values that differentiate O$_3$ formation regimes. In this study, the

thresholds of the indicators were derived based on the association of the O$_3$ reduction relative to the baseline emission scenario resulting from NO$_x$ or VOC emission reductions (i.e., ΔO$_{3\ NOx}$ or ΔO$_{3\ VOC}$), with concurrent indicator values at all grid cells in Jiangsu Province. Therefore, two additional runs with either a 35% decrease in NO$_x$ or a 35% decrease in VOC emissions were performed following Sillman et al. (1998) and other studies (Xie et al., 2014; Peng et al., 2011). The resulting O$_3$ changes (ΔO$_{3\ NOx}$ and ΔO$_{3\ VOC}$) were obtained to identify the O$_3$ formation regimes at each grid cell, with the

criteria given in Table 1. It should be noted that the average changes in O$_3$ during 1–4 p.m. during the simulation period were used when O$_3$ peaked and was of most concern.

**Table 1: Identification of the O$_3$ formation regimes at each grid cell based on the O$_3$ changes resulting from perturbed emissions as in Sillman et al. (1998) or the sensitivity coefficients using the high-order decoupled direct method (HDDM) as in Wang et al.**

**(2011).**

| Method | O$_3$ formation regime | Definition |
|---|---|---|
| Perturbed simulations | VOC-limited | ΔO$_{3\ VOC}$ ≥ 5 ppb and (ΔO$_{3\ VOC}$ − ΔO$_{3\ NOx}$) ≥ 5 ppb |
| | NO$_x$-limited | ΔO$_{3\ NOx}$ ≥ 5 ppb and (ΔO$_{3\ NOx}$ − ΔO$_{3\ VOC}$) ≥ 5 ppb |
| High-order decoupled direct method (HDDM) | VOC-limited | $S^1_{VOCs}$ ≥ 5 ppb and ($S^1_{VOCs}$ − $S^1_{NOx}$) ≥ 5 ppb |
| | NO$_x$-limited | $S^1_{NOx}$ ≥ 5ppb and ($S^1_{NOx}$ − $S^1_{VOCs}$) ≥ 5 ppb |

This study focused on four photochemical indicators that are frequently used, including P$_{H2O2}$/P$_{HNO3}$, HCHO/NO$_2$, HCHO/NO$_y$, and NO$_y$, with the underlying mechanisms extensively described in the literature (Xie et al., 2014; Kwok et al., 2015; Sillman, 1995; Duncan et al., 2010). The indicator values for each grid cell were calculated based on the simulated

concentrations of related species, except for P$_{H2O2}$/P$_{HNO3}$, which were obtained using the integrated reaction rate (IRR) of the CMAQ process analysis tool (Gipson and Young, 1999). The percentile distribution of the indicator values for the VOC-





limited grid cells and the NO$_x$-limited grid cells were examined individually. The photochemical indicators typically showed higher values at NO$_x$-limited grid cells than in VOC-limited locations. Finally, the thresholds of a certain indicator were determined using the 95th percentile value of the indicator for the VOC-limited grid cells and the 5th percentile value for the

NO$_x$-limited grid cells as the lower and upper limits of the transition intervals, respectively. Indicator values lower than the transition intervals suggested O$_3$ formation was VOC-limited, while higher values than the transition intervals were associated with NO$_x$-limited regimes. In some cases, the 95th percentile VOC-limited value was higher than the 5th percentile NO$_x$-limited value, indicating the indicator was invalid and could not be used to determine the O$_3$ sensitivity properly.

**2.4 Evaluation of the indicators**

**2.4.1 Evaluation metrics**

Some metrics have been developed to estimate the uncertainties in the determination of O$_3$ regimes using photochemical indicators (Wang et al., 2011; Ye et al., 2016). This work applied Error A, Error B, and the overall accuracy (OA) in Ye et al. (2016) for the evaluation, with the equations given in Table 2. For example, the ErrA_NO$_x$ describes the situation where

among all grid cells (colored areas in Fig. 2d) with indicator values higher than the upper limit of the transition interval (i.e., assigned to NO$_x$-limited regimes with the indicator), there existed VOC-limited grid cells that were incorrectly assigned to the NO$_x$-limited regime. ErrB_NO$_x$ estimates the fraction of the NO$_x$-limited grid cells (blue areas in Fig. 2d) that are not classified as a NO$_x$-limited regime since the corresponding indicator values are lower than the upper threshold. ErrA_VOC and ErrB_VOC were defined analogously. OA describes the fraction of all correctly classified grid cells using the indicator at

all VOC- or NO$_x$-limited grid cells.

**Table 2: Metrics of the indicator performance evaluation provided in Ye et al. (2016).**

| Metric | Equation |
|---|---|
| Error A_NO$_x$ | $\dfrac{\text{Number of VOC-limited grid cells with indicator values above the upper threshold}}{\text{Number of all NO}_x\text{-limited and VOC-limited grid cells with indicator values above the upper threshold}} \times 100\%$ |
| Error B_NO$_x$ | $\dfrac{\text{Number of NO}_x\text{-limited grid cells with indicator values below the upper threshold}}{\text{Number of all NO}_x\text{-limited grid cells}} \times 100\%$ |
| Error A_VOC | $\dfrac{\text{Number of NO}_x\text{-limited grid cells with indicator values below the lower threshold}}{\text{Number of all NO}_x\text{-limited and VOC-limited grid cells with indicator values below the lower threshold}} \times 100\%$ |
| Error B_VOC | $\dfrac{\text{Number of VOC-limited grid cells with indicator values above the lower threshold}}{\text{Number of all VOC-limited grid cells}} \times 100\%$ |
| OA | $OA=(\dfrac{\text{Number of VOC-limited grid cells with indicator values below the lower threshold} + \text{Number of NO}_x\text{-limited grid cells with indicator values above the upper th}}{\text{Number of all NO}_x\text{-limited and VOC-limited grid cells}}$ |





The indicator evaluation was conducted for the simulation in 2017 (see section 2.1). Additionally, a simulation for the same period (over July 22–31) in 2018, which has an identical model configuration and inputs in all aspects except for the meteorology, was performed for the evaluation as well. Thus, applications of the photochemical indicators with derived thresholds in a different location within the region, in a different year, or with changes in both, can be examined separately. This provides insight into the uncertainties associated with emissions or meteorology.

### 2.4.2 O$_3$ isopleths

The O$_3$-NO$_x$-VOC sensitivity determined by photochemical indicators was compared with the O$_3$ isopleths that describe the nonlinear relationship between O$_3$ and the precursors and inform decision-makers of emission control strategies in a different manner. The response of O$_3$ to the domain-wide reductions in VOC and NO$_x$ emissions at three representative sites (CCM, ZHM, and LS) during 1–4 p.m. during the simulation period was investigated. Specifically, a total of 36 emission scenarios with anthropogenic VOC and NO$_x$ emissions reduced by 0, 20, 40, 60, 80, and 100%, both singly and in combination were simulated to construct the O$_3$ isopleth diagram. The O$_3$ formation regimes based on the thresholds derived in section 2.3 are indicated with color and overlapped with the isopleths.

### 2.5 Uncertainties in the thresholds

In section 2.3, the O$_3$ formation regimes were presumably identified according to changes in O$_3$ with the emission reductions, while the rules could be different in detail. For example, Liang et al. (2006) examined changes in the 8-hr O$_3$ with a 25% reduction in VOC or NO$_x$ emissions instead of the 1-hr O$_3$ with 35% reductions, as in Sillman (1995). Accordingly, the criteria changed from 5 ppb to 2.5 ppb. Zhang et al. (2020) explored the correlation of indicator values with the relative O$_3$ changes resulting from a 50% reduction in the NO$_x$ or VOC emissions. Du et al. (2022) applied the decoupled direct method (DDM) to investigate the sensitivity of O$_3$ to precursors, instead of perturbing emissions. The impacts on the derived thresholds of the indicators were explored by conducting a few individual tests:

(1) The criteria for the O$_3$ reduction (see Table 1) were set at 2 ppb, 3 ppb, and 6 ppb.

(2) The relative changes in O$_3$ were examined using the criteria of 2%, 5%, and 8% instead of an absolute change of 5 ppb.

(3) The 35% emission reduction scenarios were replaced with a 20% and a 40% reduction in the VOC or NO$_x$ emissions.

(4) The high-order decoupled direct method (HDDM) (Cohan et al., 2005) was adopted to split the O$_3$ formation regimes in two manners referred to as "DDM1" and "DDM2". The HDDM estimated the first- and second-order sensitivity coefficients of O$_3$ to NO$_x$ and VOCs (i.e., $S_{NOx}^1$, $S_{VOCs}^1$, $S_{NOx}^2$, $S_{VOCs}^2$). With the DDM1 approach, the NO$_x$-limited locations were defined as those grid cells where $S_{NOx}^1$ was higher than 5 ppb and at least 5 ppb higher than $S_{VOCs}^1$ (Table 1), following Wang et al. (2011). The VOC-limited grid cells were defined similarly. The DDM2 approach estimated O$_3$ changes with 35% reductions in



emissions via the Taylor series expansions (Cohan et al., 2005) instead of the perturbed simulations.

## 3 Results and discussion

### 3.1 Threshold values of the photochemical indicators

The changes in the $O_3$ concentrations when reducing VOC or $NO_x$ emissions were used to determine the $O_3$ formation regimes. Figure 2a shows that high $O_3$ concentrations (averaged from 1 to 4 p.m. during July 22–31) primarily occurred along the Yangtze River and southern Jiangsu, and $O_3$ in northern Jiangsu was relatively low. The $NO_x$ emission reduction predominantly led to $O_3$ decreases in Jiangsu, while only a few locations, such as near the estuary of the Yangtze River, showed an increased $O_3$ relative to the base case (Fig. 2b). The VOCs emission reductions resulted in less significant $O_3$ decreases compared to the $NO_x$ reduction scenario (Fig. 2c). The $O_3$ sensitivity regime for each grid cell in Jiangsu was identified with the method depicted in section 2.3 (Fig. 2d). The $O_3$ formation in central and southern Jiangsu, where $O_3$ is abundant, is typically $NO_x$-limited as a result of a relatively low $NO_x$ concentration in the afternoon (Duncan et al., 2010; Jin et al., 2017; Jin and Holloway, 2015). Some areas along (and on) the Yangtze River were characterized to be VOC-limited regimes, with substantial emissions of $NO_x$ from ship emissions and industry (Sheng et al., 2022). However, not all grid cells in Jiangsu were assigned to the VOC- or $NO_x$-limited regimes. The white areas in Jiangsu in Fig. 2d indicate that $O_3$ was insensitive to both $NO_x$ and VOCs (i.e., $\Delta O_{3\ VOC} < 5$ ppb and $\Delta O_{3\ NOx} < 5$ ppb); alternatively, $O_3$ was sensitive to $NO_x$ (VOC) emissions but showed small changes in $O_3$ with the perturbation of $NO_x$ (VOC) relative to the VOC ($NO_x$) reduction scenario (i.e., $|\Delta O_{3\ VOC} - \Delta O_{3\ NOx}| < 5$ ppb).

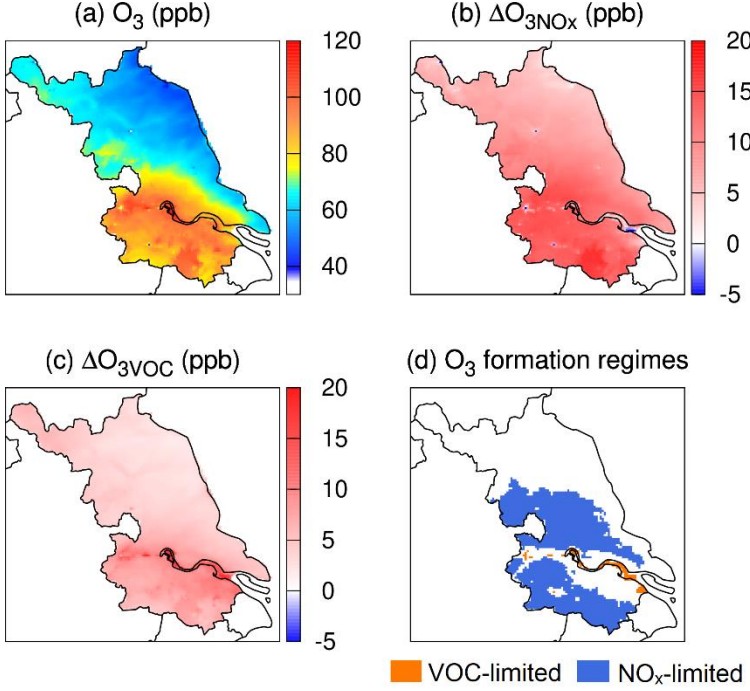

**Figure 2: (a) Simulated spatial distribution of O₃ averaged from 1 to 4 p.m. during July 22-31 in Jiangsu Province, (b-c) O₃ reduction due to a 35% reduction in domain-wide NOₓ and VOC emissions, respectively, (d) O₃ formation regimes determined with perturbed simulations.**

210

The O₃ reductions ($\Delta O_{3\ NOx}$ or $\Delta O_{3\ VOC}$) against photochemical indicator values at all grid cells over the domain were examined (Fig. 3). The mean values of the $\Delta O_{3\ NOx}$ slightly increased with higher $P_{H2O2}/P_{HNO3}$, while the $\Delta O_{3\ VOC}$ values showed a decreasing trend, indicating that a reduction in NOₓ (VOC) emissions became more (less) effective for O₃ abatement in the locations with higher indicator values. Other indicators displayed a similar pattern except for NOᵧ, which displayed an opposite result. Due to the contrasting dependence of the $\Delta O_{3\ NOx}$ and the $\Delta O_{3\ VOC}$ on the indicator values, the thresholds of the photochemical indicators that separate the O₃ formation regimes were determined (Table 3 and vertical lines in Fig. 3). The localized thresholds for $P_{H2O2}/P_{HNO3}$ (0.55, 0.75) were remarkably higher than that implemented in the model, i.e., 0.35, which would lead to misattribution of the VOC-limited or the transitional regime to the NOₓ-limited regime in some locations if the single value was directly used. The thresholds differed from those proposed by researchers in other countries, and some regions in China had significantly distinct thresholds as well (Du et al., 2022; Zhang et al., 2020; Tonnesen and Dennis, 2000a; Liu et al., 2010).



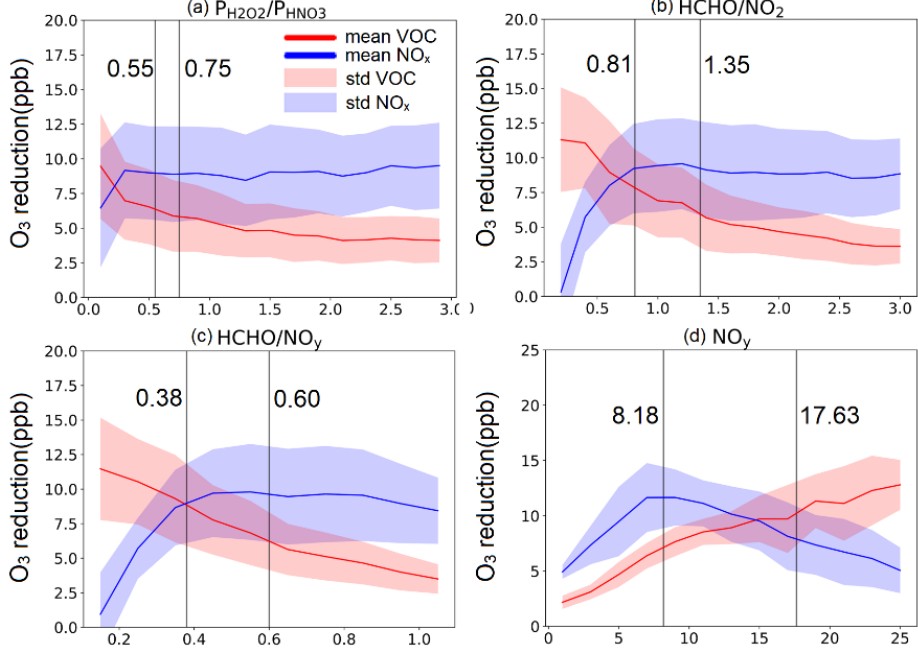

**Figure 3: Relationships between the simulated O₃ reductions with the perturbed emissions and concurrent values of $P_{H2O2}/P_{HNO3}$,**
**HCHO/NO₂, HCHO/NO$_y$, and NO$_y$ at all the grid cells in the domain. The red (blue) lines represent the average O₃ changes**
**resulting from VOC (NO$_x$) emission reductions, with the shaded areas for the standard deviations. The thresholds derived in this**
**study are indicated as the grey vertical lines.**

225

**Table 3: Thresholds of the indicators derived in this study and reported in previous work.**

| Indicators | This study | Other studies |
|---|---|---|
| $P_{H2O2}/P_{HNO3}$ | (0.55, 0.75) | US: 0.06 (Tonnesen and Dennis, 2000a) |
| | | China: 0.2 (Liu et al., 2010) |
| | | North China Plain: (0.08, 0.2) (Zhang et al., 2020) |
| | | East China: (0.30, 1.10) (Du et al., 2022) |
| HCHO/NO₂ (surface) | (0.81, 1.35) | US: (0.8, 1.8) (Tonnesen and Dennis, 2000b) |
| | | East Asia: (0.5, 0.8) (Jin et al., 2017) |
| | | East China: (0.36, 0.45) (Du et al., 2022) |
| | | Yangtze River Delta: (0.55, 1.0) (Liu et al., 2021) |
| HCHO/NO$_y$ | (0.38, 0.60) | US: 0.28 (Sillman, 1995) |
| | | California: (0.5, 0.9) (Lu and Chang, 1998) |
| | | Pearl River Delta: 0.41 (Ye et al., 2016) |



| NO$_y$ | (8.18, 17.63) | US: 20 (Sillman, 1995) |
|---|---|---|
| | | California: 5 (Lu and Chang, 1998) |
| | | Germany: 7.78 (Vogel et al., 1999) |
| | | Mexico City: 8.75 (Torres-Jardon et al., 2009) |

230

This work revisited other photochemical indicators including the surface HCHO/NO$_2$, HCHO/NO$_y$, and NO$_y$. The transition interval of HCHO/NO$_2$ was determined as (0.81, 1.35), showing a higher and wider range than in other work (Jin et al., 2017; Liu et al., 2021; Du et al., 2022), but relatively consistent with Tonnesen and Dennis (2000b). It is worth noting that the values mentioned above are all based on measurements or simulations at the ground level, while column HCHO/NO$_2$ retrieved from satellite measurements are more widely used, with thresholds of (1, 2) (Jin and Holloway, 2015; Jin et al., 2017; Duncan et al., 2010; Wang et al., 2021). The transition range for the column HCHO/NO$_2$ is typically broader than that for the surface ratio, e.g., the former of (1.17, 2.42) versus the latter of (0.36, 0.45) in East China (Du et al., 2022). The thresholds of HCHO/NO$_y$ and NO$_y$ in this study were (0.38, 0.60) and (8.18, 17.63), respectively, comparable to the values proposed previously. The comparison shown in Table 3 suggests that the thresholds were mostly location-specific, possibly as they are dependent on atmospheric conditions and emission features. The feasibility of a given threshold for an indicator applied throughout a region, or over a short-term period (the order of 1−2 years) is evaluated in section 3.2. However, the methodology that is used to derive thresholds can also lead to distinct threshold values, and this is discussed in section 3.4.

The spatial distributions of the O$_3$ formation regimes identified by the photochemical indicators with localized thresholds are exhibited in Fig. 4. Most of the areas in Jiangsu (> 80% of the grid cells covering Jiangsu, Table S2) were attributed to NO$_x$-limited regimes in terms of the summertime O$_3$ formation in the afternoon using all indicators. A few urban centers and some locations near the Yangtze River were identified as VOC-limited or transitional regimes where the four indicators largely showed differences. The O$_3$ formations in these areas were primarily VOC-limited according to P$_{H2O2}$/P$_{HNO3}$ and showed a transition from VOC-limited to NO$_x$-limited at the surrounding grid cells. However, other indicators tended to assign the areas along and to the south of the river to transitional regimes. The fraction of the VOC-limited regimes in the area in Jiangsu with P$_{H2O2}$/P$_{HNO3}$ (~13%) was considerably higher than that using HCHO/NO$_2$ and HCHO/NO$_y$ (4-5%). NO$_y$, which is the least impacted by VOC emissions, attributed the least areas to the VOC-limited regimes (~2%) concerning O$_3$ formation. Accordingly, transitional regimes were the least using P$_{H2O2}$/P$_{HNO3}$, ~5% compared to 13−15% using HCHO/NO$_2$, HCHO/NO$_y$, and NO$_y$.

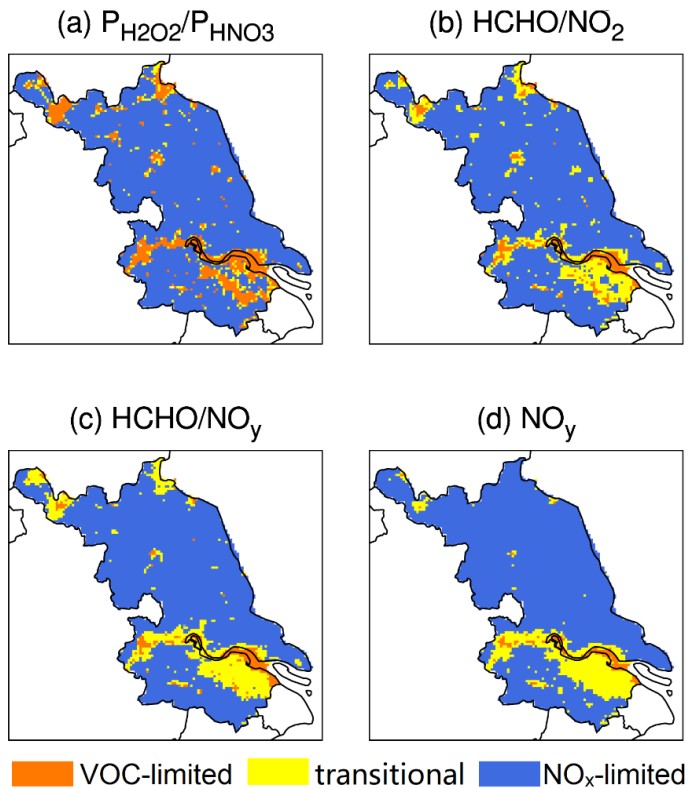

**Figure 4: O₃ formation regimes based on the indicator values and thresholds in Table 3. The orange indicates a VOC-limited regime. The blue indicates a NOₓ-limited regime. The yellow is associated with a transitional regime.**

## 3.2 Evaluation of the photochemical indicators

Table 4 shows the accuracy of the photochemical indicators in splitting the $O_3$ formation regimes with thresholds derived in this study. As expected, the ErrB and OA for the application in Jiangsu in 2017 (see the case "Sim_2017+Jiangsu" in Table 4) were approximately 5% and 95%, respectively, as a result of the nature of the method (see section 2.3). The values of ErrA were lower than ErrB, suggesting that chemical regimes determined by the indicator values were largely consistent with that based on $O_3$ changes when perturbing emissions. However, it is more likely that sensitivities indicated by $O_3$

changes do not necessarily correspond to indicator values in the correct intervals. Additionally, the ErrA_VOC and ErrA_NOₓ for $P_{H2O2}/P_{HNO3}$ were higher than that of the other indicators, likely due to less area associated with mixed $O_3$ sensitivity using $P_{H2O2}/P_{HNO3}$ (see Table S2), which could increase the chance of misclassification.





**Table 4: Evaluation of the photochemical indicators in the YRD (unit of %).**

| | | ErrA_VOCs | ErrB_VOCs | ErrA_NOx | ErrB_NOx | OA |
|---|---|---|---|---|---|---|
| Sim_2017+Jiangsu | $P_{H2O2}/P_{HNO3}$ | 3.9 | 6.6 | 0.1 | 5.1 | 94.8 |
| | HCHO/NO$_2$ | 2.9 | 5.7 | 0.0 | 5.1 | 94.9 |
| | HCHO/NO$_y$ | 0.0 | 6.6 | 0.0 | 4.7 | 95.2 |
| | NO$_y$ | 0.0 | 5.7 | 0.0 | 4.9 | 95.1 |
| Sim_2017+Other | $P_{H2O2}/P_{HNO3}$ | 7.4 | 0.0 | 0.0 | 2.0 | 98.0 |
| | HCHO/NO$_2$ | 2.2 | 0.0 | 0.0 | 1.7 | 98.3 |
| | HCHO/NO$_y$ | 2.1 | 0.0 | 0.0 | 0.7 | 99.3 |
| | NO$_y$ | 0.0 | 12.8 | 0.0 | 1.3 | 98.6 |
| Sim_2018+Jiangsu | $P_{H2O2}/P_{HNO3}$ | 2.4 | 0.0 | 0.0 | 4.3 | 96.9 |
| | HCHO/NO$_2$ | 0.0 | 0.8 | 0.0 | 7.0 | 94.8 |
| | HCHO/NO$_y$ | 0.0 | 0.8 | 0.0 | 10.1 | 92.2 |
| | NO$_y$ | 0.0 | 8.3 | 0.3 | 0.3 | 97.4 |
| Sim_2018+Other | $P_{H2O2}/P_{HNO3}$ | 2.7 | 7.0 | 0.1 | 1.7 | 98.2 |
| | HCHO/NO$_2$ | 6.2 | 14.3 | 0.1 | 1.6 | 98.0 |
| | HCHO/NO$_y$ | 0.5 | 12.2 | 0.1 | 1.2 | 98.5 |
| | NO$_y$ | 0.0 | 51.3 | 0.1 | 1.3 | 97.3 |

The above evaluation also showed that emissions could have a minor effect on the performance of the indicators on a regional scale. The thresholds derived from the simulation in Jiangsu well identified O$_3$ sensitivities in other regions of the YRD (see the case "Sim_2017+Other"), where emissions were significantly different (e.g., abundant biogenic VOCs in the southern YRD vs. dominance of anthropogenic emissions in Jiangsu) and with OAs greater than 98% for all the indicators. This suggests that it is plausible to specify regionwide thresholds for photochemical indicators. In addition, the indicators can generally separate O$_3$ formation regimes in Jiangsu for the same period in 2018 (see the case "Sim_2018+Jiangsu") when the meteorology could differ from that in 2017. However, HCHO/NO$_y$ and HCHO/NO$_2$ appeared to be more easily affected by meteorology, as indicated by the ErrB_NO$_x$ that showed that many NO$_x$-limited grids had indicator values lower than the upper thresholds. The OAs were relatively high in the case when both the emissions and meteorology changed (see the case "Sim_2018+Other"), although the ErrB_VOC were remarkably higher. This was due to fewer VOC-limited grids in the YRD, half of which had indicator values within the transition intervals and thus were misclassified as the transitional regime. The evaluation showed that indicators could perform better in detecting O$_3$ formation regimes that were predominant. However, the drawback of the evaluation is that it does not account for grids that are neither VOC-limited nor NO$_x$-limited in terms of the O$_3$ formation based on the $\Delta O_3 \,_{NOx}$ or the $\Delta O_3 \,_{VOC}$ (i.e., the white area in Jiangsu in Fig. 2d), and





the uncertainties of the photochemical indicators in these areas are unknown.

### 3.3 Consistency with the $O_3$ isopleths

$O_3$ formation regimes identified by the indicator values with varied combinations of VOC and $NO_x$ emissions were plotted

together with the $O_3$ isopleths to examine the indicators for implications in effective emission control strategies. Figure 5 and Figs. S3–S4 show the comparison at the CCM, ZHM, and LS sites, respectively. As no significant variations were found at the three sites, only the plot for CCM is shown in the main text. The $P_{H2O2}/P_{HNO3}$ value with the base emission points to a VOC-limited regime at the CCM site, whereas the isopleths suggest that a reduction in the $NO_x$ emissions should lead to reduced $O_3$. This also happens in some cases with lower VOC and $NO_x$ emissions, indicating that $P_{H2O2}/P_{HNO3}$ is likely to

underestimate the positive sensitivity of $O_3$ to $NO_x$. The indicators were assumed to have constant thresholds that do not change with emissions; this could partially explain the inconsistency of the indicators with the $O_3$ isopleths. The $P_{H2O2}/P_{HNO3}$ values at the LS and ZHM sites showed similar results, except that the $O_3$ formation at LS site shifted from VOC-limited to a transitional regime earlier with decreasing $NO_x$ emissions than at the other two sites as indicated by the ratio.

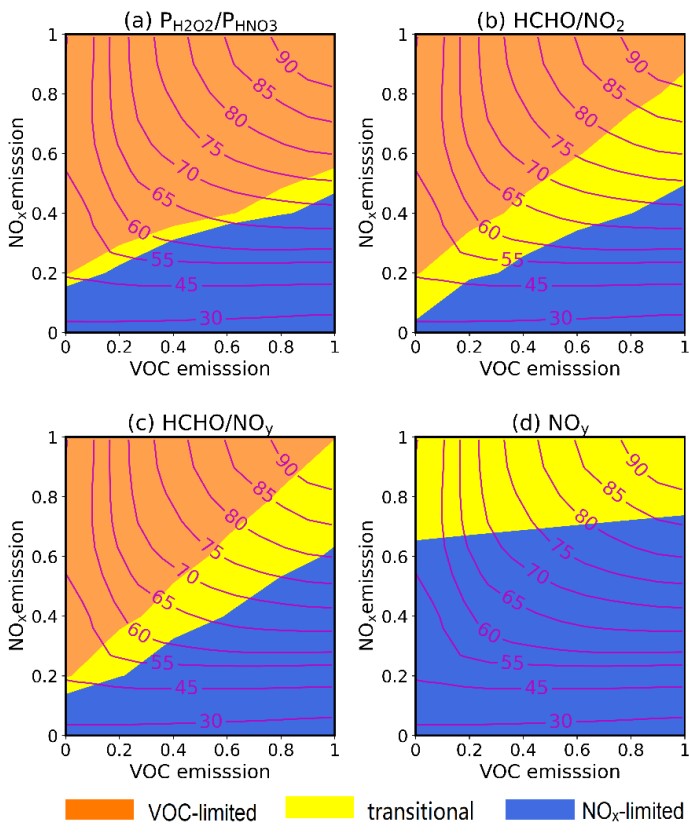



**Figure 5: O$_3$ isopleths (red lines) overlap with the O$_3$ formation regimes (shading color) identified with P$_{H2O2}$/P$_{HNO3}$ (a), HCHO/NO$_2$ (b), HCHO/NO$_y$ (c), and NO$_y$ (d) at the CCM site. The orange indicates a VOC-limited regime. The yellow indicates a transitional regime, and the blue indicates a NO$_x$-limited regime.**

It appears that HCHO/NO$_y$ was the most consistent with the O$_3$ isopleths among the four indicators, followed by HCHO/NO$_2$. Both HCHO/NO$_y$ and HCHO/NO$_2$ suggest that all three sites resided in transitional regimes under the base emission scenario, except for CCM using HCHO/NO$_2$, which showed that O$_3$ would respond to NO$_x$ changes negatively. However, HCHO/NO$_y$ and HCHO/NO$_2$ had more emission scenarios assigned to transitional regimes compared with P$_{H2O2}$/P$_{HNO3}$, which was in agreement with the spatial distributions shown in Fig. 4. This implies that the two indicators emphasize controlling both VOC and NO$_x$ rather than a single pollutant. However, the HCHO levels could be affected by primary emissions (Liu et al., 2021) or highly variable due to the dependence of isoprene on temperature (isoprene dominating HCHO production with intensive biogenic emissions) (Duncan et al., 2010). Therefore, the link between the HCHO/NO$_y$ (or HCHO/NO$_2$) values and the O$_3$-VOC-NOx sensitivity and resulting emission control strategies of anthropogenic precursors (that are controllable) is more uncertain under certain conditions.

The O$_3$ formation at the CCM site was dominated by NO$_x$-limited regimes based on NO$_y$, which was predominantly affected by NO$_x$ emissions, and did not show a sole sensitivity to VOCs (Fig. 5d). This was different from other indicators as well as the O$_3$ isopleths in terms of policy implications. The comparison between the O$_3$-VOC-NO$_x$ sensitivity revealed by NO$_y$ and the O$_3$ isopleths showed that the indicator tended to overestimate the response of O$_3$ to NO$_x$ (i.e., the VOC-limited regimes were misclassified as transitional regimes in the upper left corner of Fig. 5d) or underestimate the O$_3$-VOC sensitivity (i.e., the transitional regimes were misclassified as NO$_x$-limited regimes in the middle-left portion of Fig. 5d).

## 3.4 Uncertainties associated with the methodology

Apart from local emissions and meteorology that could partially explain the gap between the indicator thresholds proposed earlier and in this work, we found that the methodology to derive the thresholds could be slightly different and the impacts are unknown. Therefore, a series of tests were conducted to examine how those details in the methodology alter the thresholds of P$_{H2O2}$/P$_{HNO3}$ (Fig. 6).

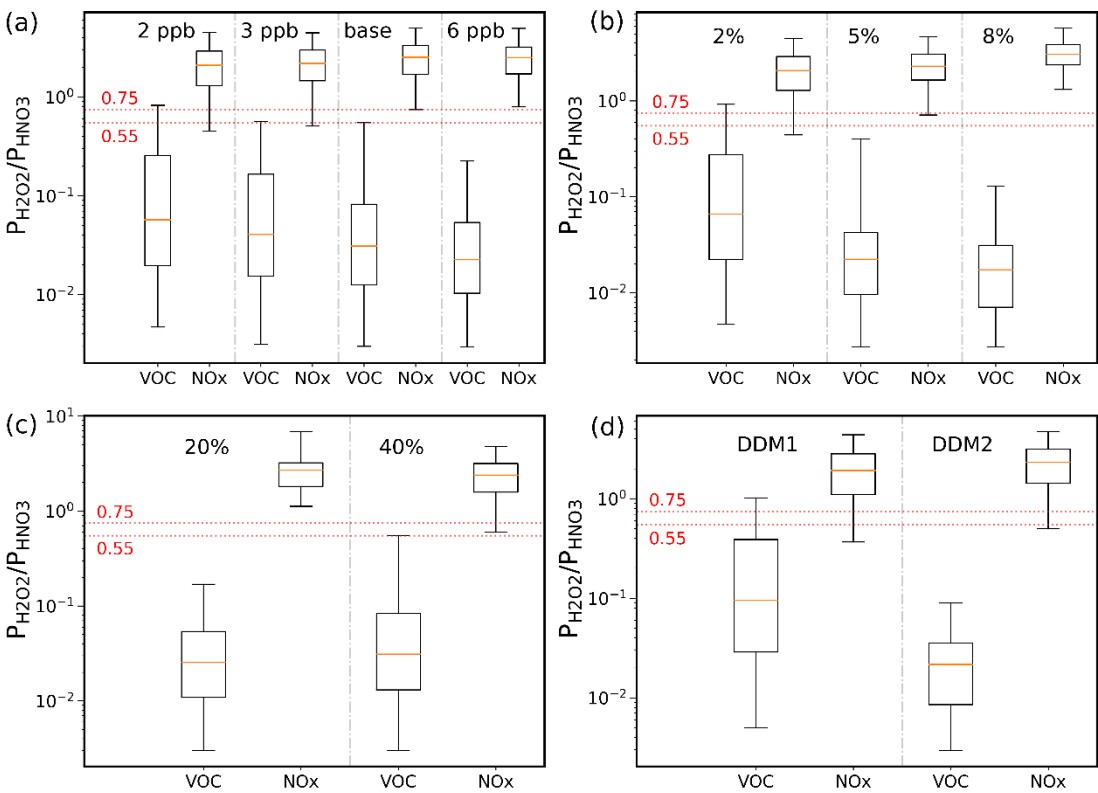

**Figure 6: The percentile distributions of the $P_{H2O2}/P_{HNO3}$ values at the VOC- or NO$_x$-limited grid cells with different setups in methodology. The boxes indicate the 25th and 75th percentiles of the indicator values with the median values marked with red horizontal lines in the boxes, and the whiskers extend to the 5th and 95th percentiles. In each group, the upper whisker of the box in the VOCs column and the lower whisker of the box in the NO$_x$ column reflect the lower and upper values of the threshold intervals, respectively. The threshold intervals in Table 3 are indicated with red dotted lines.**

First, the VOC- (NO$_x$-) limited grid cells were defined as the locations where the reduction in O$_3$ as a response to the VOC (NO$_x$) reduction exceeded the O$_3$ reduction associated with the reduced NO$_x$ (VOC) by more than 2 ppb, 3 ppb, and 6 ppb instead of the original 5 ppb (see Table 1). The threshold intervals are wider with higher criteria, e.g., the range of (0.3, 0.8) with sensitivity differences of > 6 ppb versus (0.55, 0.75) with > 5 ppb, leading to fewer VOC- or NO$_x$-limited grid cells based on the indicator and more transitional grid cells with less chance of misclassification (Fig. 6a). However, the sensitivity differences of 2−3 ppb are too small to differentiate sensitivity types of the grid cells using the indicators, as the 95th percentile of the indicator values at the VOC-limited grids is higher than the 5th percentile of the values at the NO$_x$-limited grids when determining the thresholds (see section 2.3). Hence, the indicators are invalid. Similarly, if the relative changes in O$_3$ were applied to quantify the O$_3$-VOC-NO$_x$ sensitivity, such as using the differences of 2%, 5%, or 8% as the





criteria, the derived thresholds would be considerably different as well (Fig. 6b). The relative changes make more sense since
$O_3$ changes of ~5 ppb due to a certain amount of VOC or $NO_x$ reduction are less likely to take place in locations with low $O_3$
than in relatively polluted regions. As a result, the grid cells with low $O_3$ are seemingly insensitive to precursors and were
excluded from the VOC- or $NO_x$-limited grids that were used to derive the thresholds. This can particularly affect the lower
limits of the thresholds with much fewer VOC-limited grids, as in this study, and vice versa.

In Fig. 6c, the VOC and $NO_x$ emissions were individually reduced by 20% or 40% instead of 35%, but with a constant
criterion of > 5 ppb for the sensitivity differences. The less reduction in emissions resulted in wider threshold intervals (0.18,
1.12) with a 20% reduction versus (0.55, 0.75) with a 35% reduction, and (0.55, 0.60) with a 40% reduction. This was
similar to the case with higher criteria (Fig. 6a). In addition, if the VOC- or $NO_x$-limited grids were defined based on the $S^1_{VOCs}$
and $S^1_{NOx}$ in the HDDM (see Table 1 and DDM1 in Fig. 6d), the indicator values for the VOC-limited grids and the $NO_x$-
limited grids would overlap to a large extent, with the 95th percentile of the indicator values at the VOC-limited grids of 1.0,
and the 5th percentile of the values at the $NO_x$-limited grids of 0.37. Therefore, the indicator could not be used. $S^1_{VOCs}$ and
$S^1_{NOx}$ approximate the $O_3$ changes with a 100% reduction in the VOC and $NO_x$ emissions, respectively, without considering
the nonlinearity (i.e., higher-order sensitivities) in $O_3$ chemistry. This finding is consistent with Fig. 6c which shows a quite
narrow range of threshold intervals with more reduction in the precursor emissions. Additionally, the upper and lower
thresholds derived with the DDM2 approach were 0.50 and 0.11, lower than 0.75 and 0.55, respectively. This was due to the
lower $O_3$ changes estimated with the second-order Taylor expansions than with the perturbed simulations (Fig. S5), similar to
the case with a 20% reduction in the VOC or $NO_x$ emissions, and possibly reflecting the importance of higher-order terms in
the HDDM as well. Utilizing $P_{H2O2}/P_{HNO3}$ as an example indicated that the thresholds of the photochemical indicators are
dependent on the methods or parameters in the methodology. This could be one of the major uncertainties in the application
of indicators to determine $O_3$ chemical regimes.

## 4 Conclusions

This work revisited four photochemical indicators, including $P_{H2O2}/P_{HNO3}$, $HCHO/NO_2$, $HCHO/NO_y$, and $NO_y$, in
implications of the $O_3$-$NO_x$-VOC sensitivity and emission control strategies using a case study in the YRD. The threshold
intervals for Jiangsu (one of the provinces in the YRD) were derived, (0.55, 0.75) for $P_{H2O2}/P_{HNO3}$, (0.81, 1.35) for the surface
ratio of $HCHO/NO_2$, (0.38, 0.60) for $HCHO/NO_y$, and (8.2, 17.6) for $NO_y$, based on the relationship between the simulated
indicator values and $O_3$ changes resulting from a 35% reduction in the VOC or $NO_x$ emissions. The indicators along with the
localized thresholds were applied in other areas in the YRD, for a different simulation period, and with a combination of
different locations and time to estimate the uncertainties related to the emissions and meteorology. The indicators displayed



good performance in all cases, with OA values greater than 92%, while HCHO/NO$_y$ and HCHO/NO$_2$ might have been

susceptible to meteorological factors. However, HCHO/NO$_y$ and HCHO/NO$_2$ were the most consistent with the O$_3$ isopleths among the four indicators and could be informative for policymakers. The P$_{H2O2}$/P$_{HNO3}$ ratio was less likely to attribute O$_3$ formation to mixed sensitivity and tended to underestimate the positive sensitivity of O$_3$ to NO$_x$ compared to the isopleths. In contrast, NO$_y$, which was not constrained by the VOC abundance, overestimated the positive response of O$_3$ to NO$_x$ and underestimated the O$_3$-VOC sensitivity in some conditions.


Importantly, the intrinsic characteristics of the indicators as well as the methods used to obtain the thresholds affected the effectiveness of the indicators. A series of sensitivity tests were conducted to investigate the impacts with respect to the criteria for the definition of VOC- or NO$_x$-limited grids, the amount of VOC and NO$_x$ emission perturbations, and the method, i.e., the HDDM tool versus emission reduction, to estimate the O$_3$-NO$_x$-VOC sensitivity in the first place. The

results showed that the more stringent the criteria were (i.e., with larger the O$_3$ sensitivity differences or lower emission reductions using a given criterion), the wider the transition intervals were. This led to more attribution to the mixed O$_3$ sensitivity and showed a preference for concurrent control of both precursors. However, the indicator values at the VOC- and NO$_x$- limited grids could overlap in a large part and could not split the O$_3$ sensitivity when the criteria were set too low. Finally, the first-order coefficients in the HDDM alone with a specified sensitivity difference could not be used to derive

thresholds of indicators in this study, while Taylor expansions with first- and second-order sensitivity coefficients could result in lower and broader threshold intervals.

Photochemical indicators have the advantage of identifying O$_3$ formation regimes promptly and are useful in policy implications. According to the case study in this work, we found that it was inappropriate to directly use the threshold of 0.35

for P$_{H2O2}$/P$_{HNO3}$ in the CTMs, as the value could be location-specific. More broadly, all of the indicators should be employed with specified thresholds that have been localized and thoroughly evaluated. In addition, the indicators could be different from each other. As such, it is necessary to understand the pros and cons of each indicator prior to their use.

**Code and data availability**

The CMAQ outputs are currently available upon request. All python codes used to create any of the figures are available upon request.

**Author contributions**



XL, MQ and JH conceived and designed the research. XL performed the simulations and analyzed the data. MQ, JH, HS and JL contributed to result discussion. XL and MQ wrote the manuscript with substantial contributions from all of the authors.

**Competing interests**

The authors declare that they have no conflict of interest.

**Acknowledgments**

This work was supported by the Natural Science Foundation of Jiangsu Province (BK20200815) and the National Natural Science Foundation of China (42107117, 42007187, 42021004).




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
