# Peer review of "Examining the implications of photochemical indicators on the O3-NOx-VOC sensitivity and control strategies: A case study in the Yangtze River Delta (YRD), China"

_Atmospheric Chemistry and Physics, 2022_

## Author Comment (AC1)

We thank the reviewer for the comments that improve the quality of the paper. The detailed responses are given as follows. The reviewers' comments are shown in italic font, the responses are in regular blue font, and the revised text is in blue bold font.

**Response to Referee #1**

*The values of photochemical indicators are widely used to determine the O3-NOx-VOC sensitivity with measurements, which has important policy implications. This work examined the effectiveness of four indicators such as PH2O2/PHNO3, and surface HCHO/NO2 with air quality models. It can provide decision-makers with some useful information when they use the photochemical indicators to make control strategies. The manuscript is well written and fits the scope of ACP, which is worthy of publication. However, there are a few questions as follows that need to be addressed to further improve the manuscript.*

*Major comments/questions:*

**Comment 1:** *What are the influences of regional transport on the results?*

**Response 1:** Regional transport can alter the spatial distribution of $NO_x$ and VOCs and thus influences $O_3$-precursor sensitivity. For example, Zhao et al. (2022) reported that $O_3$ production in the YRD tended to be $NO_x$-limited when source contribution from Zhejiang was elevated, which is mainly associated with the regional transport of VOCs from Zhejiang. Accordingly. the indicator ($HO_2/OH$) values increased. In contrast, $O_3$ production was VOC-limited with larger contributions from $NO_x$ emissions in Jiangsu and Anhui, and $HO_2/OH$ decreased. Therefore, we would expect that regional transport can change indicator values.

In this study, we determined $O_3$-precursor sensitivity for two simulation periods (2017 vs 2018) with different meteorology, and this can reflect the influence of regional transport on the effectiveness of photochemical indicators to some extent (see Section 3.2). As shown in Table 4, the accuracy is comparable in the two pairs of simulations, i.e., Sim_2017+Jiangsu vs Sim_2018+Jiangsu, Sim_2017+Other vs Sim_2018+Other. Although regional transport changes the values of indicators, it might have a minor influence on the performance of photochemical indicators.

**Comment 2:** *How will NH3 and inorganic aerosols such as sulfate, nitrate, and ammonium affect the HCHO/NOy and NOy?*

**Response 2:** $NH_3$ and inorganic aerosols do not affect HCHO. However, $NH_3$ can react with gaseous $HNO_3$, a major component of $NO_y$, to form particulate nitrate. Therefore, $NH_3$ affects the abundance of gaseous $NO_y$, particularly in cold seasons when the total nitrate (the sum of $HNO_3$ and particulate nitrate) predominantly resides in particles and under $NH_3$-deficient conditions. As $NO_y$ generally includes gaseous components only when it is used as a (or a part of) photochemical indicator, the $NH_3$ concentration could affect the values of HCHO/$NO_y$ and $NO_y$ under certain conditions. The YRD region is in an $NH_3$-rich environment and the formation of nitrate is insensitive to the availability of $NH_3$ (Wang et al., 2011). Moreover, $HNO_3$ dominates the total nitrate in summer (Sun et al., 2022). Therefore, the impacts of $NH_3$ and inorganic aerosols on HCHO/$NO_y$ and $NO_y$ could be negligible in this study.

*Some minor issues:*

**Comment 3:** *Page 2 line 45: ", the O3 to nitric acid ratios (O3/HNO3), and NOy)"*

**Response 3:** Thank you for your comment. We made corrections in the main text.

**Line 45: "…, the $O_3$ to nitric acid ratios ($O_3$/$HNO_3$), and $NO_y$) …"**

**Comment 4:** *Figure 1. The letters are skewed.*

**Response 4:** Thank you for your comment. We have updated Figure 1 in the main text.

**Comment 5:** *Page 6 lines 138-140: Why are the 95[th] percentile for the VOC-limited grids and the 5[th] percentile for the NOx-limited grids chosen as the boundaries of the transition interval?*

**Response 5:** We used the 5[th] and 95[th] percentiles to define the $NO_x$-VOC transition following Sillman et al. (1998) and other studies (Xie et al., 2014; Peng et al., 2011). In these practices, the 5[th] percentile $NO_x$-limited values are mostly higher than the 95[th] percentile VOC-limited values, demonstrating that $NO_x$-limited and VOC-limited chemistry is successfully distinguished. On the other hand, the percentile values can remove extremely high values in VOC-limited locations and extremely low values in $NO_x$-limited locations.

**Comment 6:** *Page 11 lines 240-242: I suggest moving this part to the methodology section.*

**Response 6:** Thank you for your comment. We have moved the sentence "The feasibility of a given threshold for an indicator applied throughout a region or over a short-term period (the order of 1–2 years) is evaluated in section 3.2." to Lines 147-148 in the methodology section. However, we keep the other sentence "However, the methodology that is used to derive thresholds can also lead to distinct threshold values, and this is discussed in section 3.4." as it is, since we want to give a possible explanation of different threshold values derived in this study compared to other studies as shown in Table 3.

**Comment 7:** *Page 14 lines 295-296: This seems to conflict with your argument that the emission had no significant influence on the thresholds of PH2O2/PHNO3.*

**Response 7:** In Section 3.2, we pointed out that emissions had no significant influence on the performance of indicators in the determination of $O_3$-precursor sensitivity. This is based on the comparison between the case "Sim_2017+Jiangsu" and "Sim_2017+Other", which show similar accuracy despite different emissions in Jiangsu and other areas in the YRD (see Table 4). Here, the thresholds derived for one area are applicable to elsewhere on a regional scale. However, we think that thresholds may change with local emissions and thus are location-specific (see Lines 239-240). This could also be the case with significant changes (up to a 100% reduction) in VOC or $NO_x$ emissions as in $O_3$ isopleth.

**Comment 8:** *The first row of Table 4 was incomplete.*

**Response 8:** Thank you for your comment. We have added names for Columns 1-2:

**Table 4:** Evaluation of the photochemical indicators in the YRD (unit of %).

| Case | Indicators | ErrA_VOC | ErrB_VOC | ErrA_NO$_x$ | ErrB_NO$_x$ | OA |
|------|-----------|----------|----------|-------------|-------------|-----|
| Sim_2017+Jiangsu | P$_{H2O2}$/P$_{HNO3}$ | 3.9 | 6.6 | 0.1 | 5.1 | 94.8 |
| | HCHO/NO$_2$ | 2.9 | 5.7 | 0.0 | 5.1 | 94.9 |
| | HCHO/NO$_y$ | 0.0 | 6.6 | 0.0 | 4.7 | 95.2 |
| | NO$_y$ | 0.0 | 5.7 | 0.0 | 4.9 | 95.1 |
| Sim_2017+Other | P$_{H2O2}$/P$_{HNO3}$ | 7.4 | 0.0 | 0.0 | 2.0 | 98.0 |
| | HCHO/NO$_2$ | 2.2 | 0.0 | 0.0 | 1.7 | 98.3 |
| | HCHO/NO$_y$ | 2.1 | 0.0 | 0.0 | 0.7 | 99.3 |
| | NO$_y$ | 0.0 | 12.8 | 0.0 | 1.3 | 98.6 |

| | | | | | | |
|---|---|---|---|---|---|---|
| Sim_2018+Jiangsu | $P_{H2O2}/P_{HNO3}$ | 2.4 | 0.0 | 0.0 | 4.3 | 96.9 |
| | $HCHO/NO_2$ | 0.0 | 0.8 | 0.0 | 7.0 | 94.8 |
| | $HCHO/NO_y$ | 0.0 | 0.8 | 0.0 | 10.1 | 92.2 |
| | $NO_y$ | 0.0 | 8.3 | 0.3 | 0.3 | 97.4 |
| Sim_2018+Other | $P_{H2O2}/P_{HNO3}$ | 2.7 | 7.0 | 0.1 | 1.7 | 98.2 |
| | $HCHO/NO_2$ | 6.2 | 14.3 | 0.1 | 1.6 | 98.0 |
| | $HCHO/NO_y$ | 0.5 | 12.2 | 0.1 | 1.2 | 98.5 |
| | $NO_y$ | 0.0 | 51.3 | 0.1 | 1.3 | 97.3 |

**Comment 9:** *Lines 450, 491, 521: The format of journal names should be consistent.*

**Response 9:** Thank you for your comment. We have updated the reference section.

**References**

Peng, Y.-P., Chen, K.-S., Wang, H.-K., Lai, C.-H., Lin, M.-H., and Lee, C.-H.: Applying model simulation and photochemical indicators to evaluate ozone sensitivity in southern Taiwan, Journal of Environmental Sciences, 23, 790-797, 10.1016/s1001-0742(10)60479-2, 2011.

Sillman, S., He, D., Pippin, M. R., Daum, P. H., Imre, D. G., Kleinman, L. I., Lee, J. H., and Weinstein-Lloyd, J.: Model correlations for ozone, reactive nitrogen, and peroxides for Nashville in comparison with measurements: Implications for O3-NOx-hydrocarbon chemistry, Journal of Geophysical Research: Atmospheres, 103, 22629-22644, 1998.

Sun, J., Qin, M., Xie, X., Fu, W., Qin, Y., Sheng, L., Li, L., Li, J., Sulaymon, I. D., Jiang, L., Huang, L., Yu, X., and Hu, J.: Seasonal modeling analysis of nitrate formation pathways in Yangtze River Delta region, China, Atmos. Chem. Phys. Discuss., 2022, 1-37, https://doi.org/10.5194/acp-2022-426, 2022.

Wang, S., Xing, J., Jang, C., Zhu, Y., Fu, J. S., and Hao, J.: Impact Assessment of Ammonia Emissions on Inorganic Aerosols in East China Using Response Surface Modeling Technique, Environmental Science & Technology, 45, 9293-9300, 10.1021/es2022347, 2011.

Xie, M., Zhu, K., Wang, T., Yang, H., Zhuang, B., Li, S., Li, M., Zhu, X., and Ouyang, Y.: Application of photochemical indicators to evaluate ozone nonlinear chemistry and pollution control countermeasure in China, Atmospheric Environment, 99, 466-473, 10.1016/j.atmosenv.2014.10.013, 2014.

Zhao, K., Wu, Y., Yuan, Z., Huang, J., Liu, X., Ma, W., Xu, D., Jiang, R., Duan, Y., Fu, Q., and Xu, W.: Understanding the underlying mechanisms governing the linkage between atmospheric oxidative capacity and ozone precursor sensitivity in the Yangtze River Delta, China: A multi-tool ensemble analysis, Environment International, 160, 107060, https://doi.org/10.1016/j.envint.2021.107060, 2022.

---

## Author Comment (AC2)

We thank the reviewer for the comments that improve the quality of the paper. The detailed responses are given as follows. The reviewers' comments are shown in italic font, the responses are in regular blue font, and the revised text is in blue bold font.

**Response to Referee #2**

*This paper presented the examination of different indicators for O3-NOx-VOC sensitivity based on the chemical transport model CMAQ results. Four indicators were tested, i.e. the ratio of the production rates of H2O2 and HNO3 (PH2O2/PHNO3), HCHO/NO2, HCHO/NOy, and reactive nitrogen concentrations (NOy) for the YRD region. This work determined and evaluated the threshold values of these indicators. Besides, the uncertainty caused by the method was also analyzed. Generally, the manuscript is well-written with a clear structure, and the analysis and discussion are scientifically sound. I recommend publication once the comments below have been addressed.*

*General comments:*

**Comment 1:** *The determination of NOx-limited and VOC-limited is changes of O3 by more than 5 ppbv if NOx and VOC emission reduction by 35% relative to the base run. This criterion is adopted from Sillman et al. (1998). However, the original analysis mainly focused on an ozone episode at the Nashville and vicinity area with relatively high O3 concentration (>80 ppbv). As indicate in Figure 2(a), the O3 concentration could match this criterion for the south part of YRD but not the North part. The relatively low O3 in the north part leads to relative low absolute change of O3 concentration when NOx or VOC emission reduce by 35%. In this case, the north part can still be attributed to NOx/VOC limited regime. It's not clear to me how the determination of threshold for different indicators depends on this classification.*

**Response 1:** We agree that the northern YRD can be attributed to $NO_x$ or VOC limited regimes, but not with $O_3$ changes of >5ppb when perturbing $NO_x$ or VOC emissions by 35%. As a result, many grids in the northern YRD were excluded when determining the thresholds of indicators (see Figure 2(d)), as the criteria ($\Delta O_3 \geq 5$ppb) are relatively high for these low-$O_3$ areas. It could induce uncertainties when applying the derived thresholds (mostly based on $O_3$ changes in the

polluted areas) to determine $O_3$ formation regimes in clean areas. Therefore, we adopted relative changes of $O_3$ instead of the absolute changes as the criteria in Section 3.4 (see Figure 6 (b)). The results show that the thresholds with $O_3$ changes ≥5%, corresponding to 2-3 pbb or larger in the northern YRD (with $O_3$ concentrations mostly in the range of 40-60 ppb), are comparable to the original values, while the relative change of 2% is too small to distinguish the $NO_x$-VOC transition. The lower limits of the thresholds were more affected due to fewer VOC-limited grids in the YRD. In general, the thresholds derived from the grids with $O_3$ changes of >5ppb are also appropriate for the northern YRD to determine $O_3$ formation regimes.

**Comment 2:** *In section 3.4, the $P_{H2O2}/P_{HNO3}$ is used for an example but also suggest to add similar discussion on HCHO/NO₂ or HCHO/NOy to address the statement that "By comparing with the $O_3$ isopleths, it was found that HCHO/NO₂ and HCHO/NOy showed the most consistency".*

**Response 2:** Thank you for your suggestion. We added Figures S6-S8 as the results of HCHO/NO₂, HCHO/NO$_y$, and NO$_y$ in supplemental materials, which were referred to in Lines 360-362:

**The results of HCHO/NO₂, HCHO/NO$_y$, and NO$_y$ (Figures S6-S8) were similar. They all indicated that the thresholds of the photochemical indicators are dependent on the methods or parameters in the methodology.**

[Figure]

**Figure S6: The percentile distributions of the HCHO/NO$_2$ values at the VOC- or NO$_x$-limited grid cells with different setups in methodology. The threshold intervals of HCHO/NO$_2$ in Table 3 are indicated with red dotted lines.**

[Figure]

**Figure S7: The percentile distributions of the HCHO/NO$_y$ values at the VOC- or NO$_x$-limited grid cells with different setups in methodology. The threshold intervals of HCHO/NO$_y$ in Table 3 are indicated with red dotted lines.**

[Figure]

**Figure S8: The percentile distributions of the $NO_y$ values at the VOC- or $NO_x$-limited grid cells with different setups in methodology. The threshold intervals of $NO_y$ in Table 3 are indicated with red dotted lines.**

**Comment 3:** *The overall accuracy values of NOy in some cases are higher than other photochemical indicators as shown in Table.4, however in the section 3.3, the indicator was not recommended. Please explain the discrepancy between the result mentioned above.*

**Response 3:** Table 4 and Section 3.3 show the evaluation of the indicators from two aspects. Table 4 shows the accuracy of the photochemical indicators with the base emissions in the YRD. However, in Section 3.3, with significant emission changes, $NO_y$ is substantially inconsistent with $O_3$ isopleths, particularly with VOC emission reductions. The former is based on statistics throughout the entire YRD in one emission scenario, while the latter emphasizes the comparison with $O_3$ isopleths with emission perturbations in one location.

*Specific comments:*

**Comment 4:** *The language needs improved. For example, the tense of one paragraph should be consistent. I only list a few and suggest the authors to carefully go through the paper. Line 14: examines to examined, Line 49: is VOC-limited to was VOC-limited…*

**Response 4:** Thank you for your comment. We have checked the tense and made the following revisions:

**Line 14: This study examined …**

**Line 49: … was VOC-limited in the morning …**

**Line 71: this work aimed to …**

**Line 112: Figure S2 shows that …**

**Line 151: …were not classified…were lower than…**

**Line 161: This provided …**

**Line 168: were indicated with …**

**Comment 5:** *Line 48: Explain "NOz" when it appeared for the first time.*

**Response 5:** Thanks for pointing this out. $NO_z$ has been defined at the first mention:

**Line 48: "…presented hourly $H_2O_2/NO_z$ ($NO_z = NO_y - NO_x$) …"**

**Comment 6:** *Line 50: If the threshold for an indicator is varying, it seems contradict to "robust",*

**Response 6:** Thank you for your comment. In each of the listed studies, the indicator $H_2O_2/HNO_3$ can well distinguish $O_3$ formation regimes, and even shows better performance than other indicators (Xie et al., 2014; Ye et al., 2016; Peng et al., 2011). The thresholds in each study are different. As such, $H_2O_2/HNO_3$ can be regarded as a reliable indicator to identify $O_3$ sensitivity with localized thresholds, although the thresholds could vary among regions.

**Comment 7:** *Please Briefly introduce HDDM in the section of "Methods".*

**Response 7:** Thank you for your comment. We have added the introduction of the HDDM as below:

**Lines 182-183: "The HDDM can quantify the responsiveness of air pollutants to infinitesimal perturbations of a model parameter or input (e.g., an emission rate of a precursor) with sensitivity coefficients (Cohan et al., 2005)."**

**Comment 8:** *The introduction of OA expression in Table 4 was incomplete*

**Response 8:** Thanks. We have corrected it.

**Comment 9:** *Please elaborate the determinization of thresholds for different photochemical indicators in Fig.3.*

**Response 9:** Thank you for your comment. Detailed information on the determination of the thresholds is provided in Section 2.3. We clarified it in the caption of Figure 3:

**Lines 225-226: The thresholds derived in this study (see Table 3, with the method detailed in Section 2.3) are indicated as the grey vertical lines.**

**Comment 10:** *Delete the extra brackets in Fig.3(a).*

**Response 10:** Thanks. Figure 3 has been updated in the main text.

**Comment 11:** *Please elaborate the approach to distinguish the O3 formation regime with shading colors as shown in Fig.5.*

**Response 11:** Thank you for your comment. The shading colors that distinguish $O_3$ formation regimes are based on the indicator values in each emission scenario, with the thresholds given in Table 3. We clarified it in the caption of Figure 5:

**Lines 229-300: Figure 5 $O_3$ isopleths (red lines) overlap with the $O_3$ formation regimes (shading color) identified with $P_{H2O2}/P_{HNO3}$, $HCHO/NO_2$, $HCHO/NO_y$, and $NO_y$ (the thresholds given in Table 3) at the CCM site.**

**References**

Peng, Y.-P., Chen, K.-S., Wang, H.-K., Lai, C.-H., Lin, M.-H., and Lee, C.-H.: Applying model simulation and photochemical indicators to evaluate ozone sensitivity in southern Taiwan, Journal of Environmental Sciences, 23, 790-797, 10.1016/s1001-0742(10)60479-2, 2011.

Xie, M., Zhu, K., Wang, T., Yang, H., Zhuang, B., Li, S., Li, M., Zhu, X., and Ouyang, Y.: Application of photochemical indicators to evaluate ozone nonlinear chemistry and pollution control countermeasure in China, Atmospheric Environment, 99, 466-473, 10.1016/j.atmosenv.2014.10.013, 2014.

Ye, L., Wang, X., Fan, S., Chen, W., Chang, M., Zhou, S., Wu, Z., and Fan, Q.: Photochemical indicators of ozone sensitivity: application in the Pearl River Delta, China, Frontiers of Environmental Science & Engineering, 10, 1-14, 2016.